# Pulse Proteins and Their Hydrolysates: A Comprehensive Review of Their Beneficial Effects on Metabolic Syndrome and the Gut Microbiome

**DOI:** 10.3390/nu16121845

**Published:** 2024-06-12

**Authors:** Lingyu Hong, Linlin Fan, Junchao Wu, Jiaqi Yang, Dianzhi Hou, Yang Yao, Sumei Zhou

**Affiliations:** 1Beijing Advanced Innovation Center for Food Nutrition and Human Health, School of Food and Health, Beijing Technology and Business University, Beijing 100048, China; caesarhong0424@163.com (L.H.); fxxfll@163.com (L.F.); zjpjwjc@163.com (J.W.); listen.y3@outlook.com (J.Y.); houdianzhi@btbu.edu.cn (D.H.); 2Beijing Engineering and Technology Research Center of Food Additives, School of Food and Health, Beijing Technology and Business University, Beijing 100048, China; 3Key Laboratory of Green Manufacturing and Biosynthesis of Food Bioactive Substances, China General Chamber of Commerce, Beijing 100048, China; 4Key Laboratory of Grain Crop Genetic Resources Evaluation and Utilization, Ministry of Agriculture and Rural Affairs, Institute of Crop Sciences, Chinese Academy of Agricultural Sciences, Beijing 100081, China

**Keywords:** pulse protein, bioactive peptide, anti-hypertensive, anti-hyperglycemic, anti-dyslipidemic, anti-obesity, gut microbiota

## Abstract

Pulses, as an important part of the human diet, can act as a source of high-quality plant proteins. Pulse proteins and their hydrolysates have shown promising results in alleviating metabolic syndrome and modulating the gut microbiome. Their bioactivities have become a focus of research, with many new findings added in recent studies. This paper comprehensively reviews the anti-hypertension, anti-hyperglycemia, anti-dyslipidemia and anti-obesity bioactivities of pulse proteins and their hydrolysates in recent in vitro and in vivo studies, which show great potential for the prevention and treatment of metabolic syndrome. In addition, pulse proteins and their hydrolysates can regulate the gut microbiome, which in turn can have a positive impact on the treatment of metabolic syndrome. Furthermore, the beneficial effects of some pulse proteins and their hydrolysates on metabolic syndrome have been supported by clinical studies. This review might provide a reference for the application of pulse proteins and their hydrolysates in functional foods or nutritional supplements for people with metabolic syndrome.

## 1. Introduction

Metabolic syndrome (MetS), as a non-communicable disease, is showing a rapid growth trend in both developing and developed countries [1]. The global epidemic of MetS has become a significant public health problem of worldwide concern [2]. MetS is typically characterized by a complex combination of risk factors, including hypertension, hyperglycemia (impaired glucose tolerance), dyslipidemia and abdominal obesity [3], which might significantly increase the risk for cardiovascular diseases and diabetes. In addition, patients with MetS may have disturbances in the gut microbiome, including decreased beneficial microbes and increased harmful ones [4]. Such disturbances may be closely associated with the development and progression of MetS. Moreover, the gut microbiome can also affect the host’s metabolism in a variety of ways, such as participating in the metabolism of nutrients and regulating the host’s energy balance and inflammatory responses [5]. Numerous studies have demonstrated the potential role of the gut microbiome in interfering with the host’s metabolism and influencing various risk factors for MetS [6,7]. Thus, regulation of the gut microbiome has a positive effect on the prevention and treatment of MetS.

Currently, although some drugs are available to treat or alleviate MetS, most of them are constantly accompanied by undesirable side effects [8]. Noticeably, MetS is believed to be mostly caused by a high-fat or high-sugar diet and deficient physical activity. In this case, dietary interventions are deemed as the first-line treatments, considering the difficulty of persisting in physical activity. Evidently, consumption of plant proteins and their hydrolysates has shown promising preventive and therapeutic effects in ameliorating various factors contributing to MetS, including hypertension, hyperglycemia, dyslipidemia and obesity [9]. Furthermore, the constant evidence suggests that plant proteins could influence MetS risk factors by modulating the gut microbiome [10]. Among the raw material sources of plant proteins, pulses have gained significant attention due to their high protein content, bioactivities and easy availability [11].

Pulses mainly refer to the dried seeds of leguminous plants in a general sense, excluding plants used for oil extraction like soybeans and peanuts, as well as freshly harvested pulses like fresh peas [12]. Pulses are considered the second most important food source after cereals, which are widely cultivated worldwide [13]. Pulses consumed by humans primarily include peas, faba beans, chickpeas, lentils, lupins, cowpeas, pigeon peas, etc. [14]. In some cases, pulses can serve as a substitute for meat as a protein source [15]. These high-quality pulse proteins also exhibit potential for preventing and treating MetS [16]. Notably, several studies have demonstrated the effectiveness of different pulse proteins and their hydrolysates in managing hypertension, hyperglycemia, dyslipidemia and obesity. For instance, recent research indicated that pea protein reduced postprandial blood sugar levels and stimulated insulin release in healthy adults [17]. In addition, a previous study found that a high-pea protein diet prevented weight gain and further fat accumulation in experimental rats [18]. Similarly, chickpea peptides derived from chickpea protein isolates have been shown to significantly decrease serum total cholesterol, triglycerides and low-density lipoprotein (LDL) cholesterol levels in obese rats fed a high-fat diet (HFD) [19]. Furthermore, the improvement effect of gut microbiome disorder was considered to be a key pathway for pulse proteins to alleviate MetS [10,20].

This review explores recent research advancements in the utilization of these pulse proteins and their hydrolysates for improving and treating MetS, which encompasses conditions like hypertension, hyperglycemia, dyslipidemia and obesity. The role of pulse proteins and their hydrolysates in regulating the gut microbiome and contributing to alleviating MetS are also discussed. Moreover, some clinical studies conducted on pulse proteins and their hydrolysates have been emphasized. The related scientific literature was reviewed and the studies were identified using Google Scholar (https://scholor.google.com/ (accessed on 14 March 2023)), Scopus (https://www.scopus.com/ (accessed on 14 March 2023)) and Web of Science (https://www.webofscience.com/ (accessed on 14 March 2023)) databases. Combinations of several search terms, such as “pulse”, “protein”, “hydrolysate/s”, “peptides”, “bioactive”, “in vitro”, “in vivo”, “clinical trial”, “anti-hypertensive”, “anti-hyperglycemic”, “anti-dyslipidemic”, “hypolipidemic”, “anti-obesity”, “microbiota” and “microbiome”, were applied. All literature selected was published in 2018 and later. The objective of this review is to provide insight into the health benefits of the major pulse proteins and their hydrolysates in relation to alleviating MetS. We hope that this review will provide a reference for the potential application (such as in functional foods or supplements) of pulse proteins and their hydrolysates.

## 2. Pulse Proteins and Their Hydrolysates for Improving MetS

Although pulse proteins and their hydrolysates can help reduce risk factors for MetS, such as hypertension, hyperglycemia, dyslipidemia and obesity, the focus herein is on the types of pulses used to control these risk factors varies. In the following sections, recent research advancements in using pulse proteins and their hydrolysates to control different risk factors are described in detail. Table 1 and Table 2 shows the in vitro and in vivo study results of pulse proteins and their hydrolysates on MetS. Furthermore, Figure 1 presents an overview of the potential mechanisms by which pulse proteins and their hydrolysates may be effective in managing MetS.

### 2.1. Anti-Hypertension

Hypertension is a major risk factor for cardiovascular disease, often leading to serious conditions like atherosclerosis and stroke [69]. Inhibition of angiotensin-converting enzyme (ACE) activity and renin activity can effectively lower blood pressure [70]. Currently, compared with renin inhibitor drugs, ACE inhibitor drugs are more commonly used for clinical treatment [71]. Although these anti-hypertension drugs are effective, they are expensive to synthesize and have side effects [72]. People still hope to find natural active substances from food [73]. Some pulse proteins and their hydrolysates have been found to have anti-hypertension activity in vitro and in vivo studies [74]. The molecular mechanisms involved are shown in Figure 2.

#### 2.1.1. In Vitro Studies

Protein hydrolysates derived from mung bean, pea, pigeon pea, chickpea and lupin have been found to show high ACE inhibitory activity [21,22,23,24,27,28,30,31]. However, the ACE inhibitory activity of peptide fractions in protein hydrolysates depends on the peptide molecular size [75]. Ultrafiltration treatment of pulse protein hydrolysates yielded peptide fractions with different molecular weights (MWs), in which peptide fractions with small MWs tended to have stronger ACE inhibitory activities than those with large MWs. The peptide fraction from mung bean protein hydrolysates with MWs < 3 kDa (IC_50_ = 4.66 μg/mL) showed significantly stronger ACE inhibitory activity than that with 3 < MWs < 10 kDa (IC_50_ = 10.18 μg/mL) and that with MWs > 10 kDa (IC_50_ = 10.27 μg/mL) [23]. Similarly, the peptide fraction < 1 kDa extracted from mung bean protein hydrolysates digested by bromelain showed stronger ACE inhibitory activity (IC_50_ = 0.50 mg/mL) than other peptide fractions with higher MWs (IC_50_ ranging from 0.58 mg/mL to 0.68 mg/mL) [24]. This was not only true for pulse proteins hydrolysed by enzymes personally used by scholars, as commercially available pea protein hydrolysates treated with ultrafiltration yielded the peptide fraction with MWs < 3 kDa with enhanced ACE inhibitory activity when compared with the untreated peptide mixture with large MWs [27]. The same pattern occurred in the studies of pigeon pea protein hydrolysates [30]. Intriguingly, ultrasonication treatment improved the ACE inhibitory activity of lupin protein hydrolysates [21]. This ultrasound pre-treatment might be used for other pulse proteins to verify the generalizability for better access to bioactive peptides with high ACE inhibitory activity. In addition, the protein extracts from mung bean tempe fermented by *Rhizopus* sp. had ACE inhibitory activity of up to 75% [25]. The tempe protein from hydrolysed pigeon pea also exhibited high ACE inhibitory activity (71.53%) [29]. This suggested that in addition to obtaining bioactive peptides from pulse proteins, the direct development of pulses into functional anti-hypertensive foods also had great potential. Nevertheless, a direct comparison of experimental results from different papers is a challenging endeavor. This is due to the fact that the ACE inhibitory activity of each sample is contingent upon the protein extraction procedure, the resistance of the protein to proteolysis, the treatment prior to proteolysis, the potential presence of contaminants and the composition of the hydrolysates, which is related to the specificity of the hydrolase. It is equally important to consider the reaction parameters (pH, temperature, enzyme/substrate ratio and contact time of the hydrolysis reaction) when comparing the results of different studies. Therefore, a direct comparison of IC50 values or percentage inhibition is meaningless. In addition, the abovementioned studies have also demonstrated the renin inhibitory activity of protein hydrolysates of mung bean and pigeon pea [24,30]. In summary, ACE catalyzes angiotensin I to angiotensin II, the efficient vasoconstrictive effect of which increases blood pressure. Moreover, ACE deactivates bradykinin which can decrease blood pressure. The aforesaid hydrolysates of pulse proteins exert a blood pressure-lowering function by inhibiting ACE activity. Peptide fractions of different molecular weights obtained by ultrafiltration of crude protein hydrolysates tended to have stronger ACE inhibitory activity and lower IC_50_ values than crude protein hydrolysates. Among these peptide fractions, the smaller the molecular weight of the peptide fraction, the stronger its ACE inhibitory activity. Specific peptides identified from the hydrolysates, on the other hand, had more potent ACE inhibitory activity. However, there is still a lack of studies comparing the ACE inhibitory activity of multiple types of pulse protein hydrolysates simultaneously. Given the differences in composition and peptide hydrolysis processes of pulse proteins, it is believed that finding pulse protein hydrolysates with stronger activity is important to promote their application in anti-hypertensive foods.

Differently from ACE, angiotensin-converting enzyme II (ACE2) degrades Ang II to angiotensin (1–7) in order to perform the counterbalancing effects of Ang II via Mas receptors. The upregulation of ACE2 through bioactive peptides also contributes to lowering blood pressure. The peptide AKSLSDRFSY, obtained by combined hydrolysis of pea protein with thermoase and pepsin, exhibited ACE2 upregulation activity [26]. Two other peptides (LSDRFS and SDRFSY) obtained by treating this peptide using in vitro digestion methods showed comparable effects to the parent peptide in upregulating ACE2 expression in vascular smooth muscle cells [26]. These provide new strategies for the identification of anti-hypertensive peptides from pulse proteins. In future studies, identification of ACE2-upregulating peptides from pulse proteins of other types besides peas could be attempted.

#### 2.1.2. In Vivo Studies

The available in vivo studies have typically used models of spontaneously hypertensive rats (SHRs) to study the anti-hypertensive effects of pulse proteins and their hydrolysates. It was reported that rats were orally administered mung bean protein hydrolysates (MPHs) and polypeptides (LPRL, YADLVE, LRLESF, HLVHHEN and PGSGCAGTDL) with MWs < 1 kDa at a dose of 20 mg/kg body weight to evaluate their anti-hypertensive effect, showing that mung bean protein peptides had a more significant anti-hypertensive effect (up to −36 mmHg) than MPHs (up to −15 mmHg) [24]. The peptide YADLVE had a longer-lasting effect at a 24 h value of −27 mmHg. In another study, after a single oral administration of pigeon pea protein isolate and hydrolysates (100 mg/kg body weight), pepsin-pancreatin hydrolysed pigeon pea protein sustained a 24 h low systolic blood pressure (SBP) (−32.12 mmHg) better than pigeon pea protein isolate and the other hydrolysates in SHRs [52]. Moreover, processing methods had an impact on the anti-hypertensive effect of pulse protein hydrolysates. Chickpea flour was subjected to an extrusion process in an extruder (feed rate of 30 rpm and extrusion temperature of 150 °C) which altered the digestibility of the protein. Then, SHRs received (squeezed or unsqueezed) chickpea protein hydrolysates (1.2 g/kg body weight (BW)) intragastric treatment, the anti-hypertensive effect of extruded hydrolysates within 3–5 h (between −15.9 and −20.0 mmHg) was significantly better than that of unextruded hydrolysates (between −29.9 and −61.9 mmHg) [53].

In conclusion, pulse proteins and their hydrolysates had significant ACE and renin inhibitory activities, showing an in vitro and in vivo anti-hypertensive effect. In future studies, will be is necessary to find more suitable pulse proteins and peptide hydrolysis methods with higher anti-hypertensive activities.

### 2.2. Anti-Hyperglycemia

Hyperglycemia might lead to damage in various tissues and organs throughout the body and is a significant contributing factor to kidney failure, neuropathy and cardiovascular disease [76]. Quite a few studies have demonstrated that consumption of pulse proteins and their hydrolysates can enhance glucose metabolism and reduce fasting blood glucose levels [77].

#### 2.2.1. In Vitro Studies

The inhibitory ability of dipeptidyl peptidase IV (DPP-IV), α-amylase and α-glucosidase activities, at least in part, can be responsible for anti-hyperglycemia effects. Recent studies have identified protein hydrolysates of lupin and pea as having good DPP-IV inhibitory activity [27,32]. In addition to complete inhibition of DPP-IV activity, the bioactive peptides obtained by hydrolysis of γ-conglutin, a specific protein component obtained from lupin, increased glucose uptake by 6.5-fold compared to untreated cells, increased insulin receptor sensitivity and reduced gluconeogenesis by 50% [32]. Furthermore, γ-conglutin peptides did not promote insulin secretion from β-cells but triggered a potent insulin mimetic effect by activating the insulin signaling pathway responsible for glycogen, protein synthesis and glucose transport into myotubes [33].

Furthermore, protein hydrolysates of lupin, pea and pigeon pea were found to have inhibitory activities of α-amylase and α-glucosidase [21,30,35] and the inhibitory activity of lupin protein hydrolysates could be enhanced by ultrasound pre-treatment [21]. In addition, three types of peptides (SPRRF, FE and RR) were predicted to be the peptides with the strongest α-glucosidase inhibitory activities by in-silico analysis of lupin protein hydrolysates, while the other three peptides (RPR, PPGIP and LRP) were predicted to be the peptides with the strongest α-amylase inhibitory activities [34]. For chickpea protein hydrolysates, they showed only an inhibitory effect on α-glucosidase activity and no significant inhibitory effect on α-amylase activity [37]. Noticeably, chickpea peptides with good α-glucosidase inhibitory activity could be prepared using an enzyme–bacteria synergy method [38], which provided a new approach for the preparation of anti-hyperglycemic bioactive peptides. Nevertheless, cowpea proteins hydrolysed with pepsin or trypsin were reported to have good α-amylase inhibitory activity (IC_50_ values of 0.127 mg/mL and 0.223 mg/mL, respectively) [39]. In addition to inhibiting the activities of the above enzymes, pea protein-derived peptides inhibited hepatic gluconeogenesis in Alpha Mouse Liver 12 (AML-12) cells by inhibiting the cyclic adenosine monophosphate (cAMP) response element binding protein-mediated signaling pathway in the gluconeogenic pathway [36].

#### 2.2.2. In Vivo Studies

Studies on hypoglycemic effects in vivo have been conducted for pulse proteins and the associated functional factors, such as γ-conglutin from lupin protein. For HFD-induced mice (high-fat rations provided 5.1 kcal/g of energy, with protein, carbohydrate and fat accounting for 18, 22 and 60% of total dietary energy, respectively, inducing hyperglycemia and steatohepatitis), supplementation with 2 g/kg BW of lupine protein isolate for 3 weeks significantly decreased their blood glucose and improved insulin sensitivity via upregulating the hepatic Fasn gene and downregulating the Gys2 and Gsk3b genes [55]. In another study, a combination of γ-conglutin and lupanine was used to treat type 2 diabetic (T2D) rats (a rat model of spontaneous diabetes mellitus commonly used as an animal model for type II diabetes mellitus) [54]. The results of deoxyriboNucleic acid (DNA) microarray and bioinformatics analysis showed that the combination of 28 mg/kg BW γ-conglutin and 20 mg/kg BW lupanine significantly reduced blood glucose and lipid levels.

The hypoglycemic activities of protein peptides from mung bean and pea have also been confirmed by several studies. Dietary supplementation with mung bean peptides (245 mg/kg) for 5 weeks significantly reduced hyperglycemia and insulin resistance in HFD-induced prediabetic mice [20]. T2D mice treated with a combination of pea oligopeptide (3200 mg/kg BW) and metformin for 4 weeks yielded more significant reductions in blood glucose levels, lipid profiles and hepatic fat deposition than those of metformin alone [56]. This may be due to the fact that pea oligopeptides promoted the absorption of metformin or acted as a vehicle for producing synergistic effects. Furthermore, it was found that pea glycoprotein PGP2 (400 mg/kg) was effective in reducing fasting blood glucose levels, increasing glucose tolerance levels and improving insulin resistance in HFD- and streptozotocin (STZ)-induced T2D mice in a dose-dependent manner [57]. 

Overall, pulse proteins and their hydrolysates showed significant DPP-IV inhibitory activity, thus leading to increased insulin levels and lowered blood glucose. Their α-amylase and α-glucosidase inhibitory activities resulted in delayed glucose uptake and reduced postprandial blood glucose levels. Reducing insulin resistance and increasing insulin sensitivity are also the main ways in which pulse proteins and their hydrolysates achieve their anti-hyperglycemic activity. In general, the regulatory mechanism of the anti-hyperglycemic activity of pulse proteins and their hydrolysates may be a combination of the above pathways.

### 2.3. Anti-Dyslipidemia

Dyslipidemia is a common symptom of MetS, characterized by hypercholesterolemia, hypertriglyceridemia, mixed hyperlipidemia or low high-density lipoprotein-(HDL) cholesterolemia [78]. If this symptom is not controlled over a long period, it can lead to cardiovascular diseases such as atherosclerosis, myocardial infarction and stroke [79]. Recently, pulse proteins have attracted more attention due to their potential anti-dyslipidemic function [80].

#### 2.3.1. In Vitro Studies

There are two sources of cholesterol in human cells: one is the ability to use acetyl coenzyme A (acetyl-CoA) to synthesize cholesterol; the other is that low-density lipoprotein (LDL) in plasma can bind to the low-density lipoprotein receptor (LDLR) on the cell membrane and enter the cell through cytotransmission, after which the LDL is hydrolysed in the lysosome to release cholesterol. These mechanisms are shown in Figure 3. Inhibiting the activity of 3-hydroxy-3-methylglutaryl CoA reductase (HMG-CoAR), which is the rate-limiting enzyme of the cellular cholesterol biosynthesis pathway, can reduce cholesterol synthesis and blood cholesterol levels. A variety of cholesterol-lowering peptides from lupin and chickpea proteins with HMG-CoAR inhibitory activity have been identified. The peptide YDFYPSSTKDQQS obtained from lupin protein hydrolysates could significantly inhibit the HMG-CoAR activity, thus enhancing the ability of human hepatocellular carcinomas (HepG2) cells to take up extracellular low-density lipoproteins [40]. Another lupin peptide GQEQSHQDEGVIVR (T9) also exhibited significant inhibitory activity (IC_50_ = 99.5 ± 0.56 µM) [43]. Similarly, VFVRN, a kind of chickpea peptide, showed the ability to inhibit the total triglyceride (TC) synthesis by decreasing the expressions of HMG-CoAR in HepG2 cells [19]. Moreover, hydrolysates of mung bean, pigeon pea, cowpea and faba bean proteins also showed HMG-CoAR inhibitory activity [45,46,47,48,49].

Proprotein convertase (PC) subtilisin/kexin type 9 (PCSK9) inhibition is a new approach to lowering cholesterol [81]. A new lupin protein peptide, P5-Best (LYLPKHSDRD), was reported to show good PCSK9 (IC_50_ = 0.7 µM) and HMG-CoAR (IC_50_ = 88.9 µM) inhibitory activities [44]. PCSK9^D374Y^, a mutation of PCSK9, was capable of increasing the risk of hypercholesterolemia [82]. T9, the lupin peptide mentioned above, could also impair the protein–protein interaction between PCSK9^D374Y^ and LDLR (IC_50_ = 285.6 ± 2.46 μM) to elevate the protein level of LDLR, thus restoring the uptake of external LDL by HepG2 cells (83.1 ± 1.6%) [41]. Furthermore, the T9 analogue (GQRQWKQAEGVMVR) could improve the LDLR expression on the liver cell surface by 84% at the concentration of 10 μM [42]. It is necessary to further investigate the anti-dyslipidemic mechanisms and effects of T9 and its analogues.

In in vitro lipid-lowering studies, the inhibition of cholesterol’s micellar solubility is also considered one of the mechanisms for lowering cholesterol. Hydrolysates of mung bean, pigeon pea and faba bean proteins have been found to have this lipid-lowering effect [45,46,49]. Noticeably, thermosonication pre-treatment could improve the inhibition of cholesterol solubilization into micelles using mung bean protein hydrolysates [45]. In another study, thermal pre-processing of faba bean protein also resulted in peptides with higher inhibition [49].

#### 2.3.2. In Vivo Studies

In recent years, many lipid-lowering studies have been conducted on pulse proteins and their hydrolysates from lupin, chickpeas, pigeon beans, lentils and peas using model mouse or rat experiments. Lupin protein hydrolysates (LPHs) (100 mg/kg) orally administered for 12 weeks showed significant hypocholesterolemic effects in Western diet-fed ApoE^−/−^ mice (a commonly used atherosclerotic mouse strain, serving as a spontaneous model of atherosclerosis that most resembles the atherosclerotic process in humans) through the modulation of LDL receptor (LDLR) and PCSK9 pathways [59] and further achieved the effects of alleviating oxidative stress and aortic inflammation, preventing the early stages of atherosclerosis [58]. In addition, a combination of lupin proteins (LPs) with natural macromolecules, such as chitosan (CH) and α-cyclodextrin (α-CD), promised a superior therapeutic effect in a hyperlipidemic rat model (acute hyperlipidemia induced by intraperitoneal injection of 1 g/kg body weight of Poloxamer 407 in nocturnally fasted rats) [60].

Two kinds of chickpea peptides (ChPs) orally administered for 6 weeks significantly alleviated hyperlipidemia by upregulating the expression of peroxisome proliferator-activated receptors (PPARs) α and LDLR and downregulating the expression of sterol regulatory element binding protein-2 (SREBP-2) in HFD-induced obese rats [19], and another chickpea peptide (RQSHFANAQP) (40 mg/kg for 4 weeks) also led to a remarkable reduction in serum total cholesterol (TC), triglyceride (TG) and hepatic TG levels in hyperlipidemic mice, accompanied by a notable increase in serum superoxide dismutase activities [62]. In hypercholesterolemic rats, both pigeon pea and lentil protein hydrolysates showed good effects in reducing the atherosclerosis index and hyperlipidemia [61,65], whereas, cowpea protein and pea protein hydrolysates exhibited excellent TC-lowering effects in hypercholesterolemic rats [63,64].

The anti-dyslipidemic ability of pulse proteins and their hydrolysates is mainly related to the inhibition of HMG-CoA activity and cholesterol micellar solubility in vitro, as well as the regulation of key genes of hepatic lipid metabolism in vivo. Among them, key signaling pathways such as PPAR and PSCK9 are mainly involved in these regulatory mechanisms. In future research, it will be necessary to normalize and standardize the dosages and intervention times of pulse proteins and their hydrolysates.

### 2.4. Anti-Obesity

Obesity, the most concerning symptom in metabolic syndrome, is a state caused by the excessive accumulation of fat, especially triglycerides [83]. It is usually caused by factors such as overeating, insufficient physical exercise and endocrine disorders. Noticeably, obesity is also an important factor leading to the development of type 2 diabetes mellitus (T2DM), atherosclerosis and cancer [84]. The abovementioned lipid-lowering effects of pulse proteins and their hydrolysates undoubtedly contribute to the fight against obesity and do not need much elaboration. There are certainly still other studies on the regulation of obesity by pulse proteins and their hydrolysates that are also accessible.

#### 2.4.1. In Vitro Studies

Pea albumin was found to inhibit lipid accumulation in mouse embryonic fibroblasts (3T3-L1) cells in vitro [51]. Subsequently, pea vicilin hydrolysates have been reported to modulate obesity-associated metabolic disorders by stimulating adipocyte differentiation through the upregulation of peroxisome proliferator-activated receptor γ (PPARγ) expression levels and ligand activity [50]. However, the anti-obesity function of pulse proteins and their hydrolysates has not been fully explored in recent studies. More research could focus on the exploration of other types of pulse proteins and their bioactive peptides with obesity-modulating effects and their possible anti-obesity mechanisms.

#### 2.4.2. In Vivo Studies

It has been shown that mung bean protein can regulate obesity by decreasing lipid accumulation [66]. Chickpea protein hydrolysate was shown to ameliorate HFD-induced obesity in mice by modulating inflammation [67]. Moreover, after oral administration of pea albumin (1.5 g/kg BW) for 8 weeks, HFD-fed mice lost body weight [51]. The mechanism was to regulate lipid metabolism by downregulating lipolysis and fatty acid oxidation-related key proteins such as FASN and downregulating lipogenesis in the body. Noticeably, pea protein intake did not prevent obesity in the offspring of obese mother rats (HFD-induced female rats were offered a diet supplemented with 25% pea protein for 4 weeks) [68], but it could selectively protect male offspring from abnormal lipid metabolism in adulthood.

Overall, the anti-obesity modulatory capacity of pulse proteins and their hydrolysates was not only related to stimulating adipocyte differentiation, but also to the regulation of lipid metabolism. Given the correlation between anti-obesity and lipid-lowering function, the regulatory mechanism of the latter can also be used as a reference for the former.

## 3. Regulation of the Gut Microbiome by Pulse Proteins and Their Hydrolysates

The gut microbiome is now recognized as one of the key factors regulating host health. Correlative studies have documented changes in the relative abundance of various gut bacteria in individuals with gastrointestinal phenotypes, including the relationship between the gut microbiome and MetS. In addition to correlation analyses, intervention studies and animal studies have not only demonstrated correlational relationships between the gut microbiome and several diseases, but have also confirmed causal relationships [85]. It has long been mentioned in some of the previously presented studies that pulse proteins and their hydrolysates can treat MetS by regulating the gut microbiome. Table 3 summarizes recent studies on the effects of pulse proteins and their hydrolysates on the gut microbiome, including the type of pulse proteins, experiment models, doses and durations, and key findings.

Limited literature has investigated the utilization of pulse proteins and their hydrolysates in treating hypertension and diabetes through modulating the gut microbiome. Feeding pea oligopeptides (50 mg/mL) to diet-induced (administration of a high-salt diet containing 5% NaCl and high-sugar water containing 20% fructose) hypertension rats for 3 weeks revealed their capacity to regulate their intestinal microenvironment, thus mitigating adverse effects on blood pressure, enhancing bacterial sugar decomposition and managing body fat levels [86]. Additionally, a HFD supplemented with mung bean peptide (245 mg/kg BW) for 5 weeks demonstrated a significant reversal of the gut microbiome imbalance in HFD-induced prediabetic mice, reducing the abundance of *Bacteroides chauvinii* and *Bacteroides anisopliae* [20]. A previous study has shown that compared to beef protein and casein in the diet, lupine protein can reduce cardiovascular risk in pigs (the three groups of pigs were fed lupin protein isolate, lean beef and casein, respectively, at a protein intake of 130–150 g per kg of diet for 4 weeks), accompanied by increased levels of *Bacteroidetes* and *Firmicutes phyla* [87].

The anti-obesity effect of pulse proteins has also been found to be related to the regulation of the gut microbiome. The anti-obesity effect of mung bean protein (containing a diet of 25%, *w*/*w*) became less significant in germ-free mice, suggesting that the function of mung bean protein needs to be mediated by the gut microbiome [66]. Similar results were observed in pea albumin (C57BL/6 male mice gavaged 1.5 g/kg BW pea albumin daily for 8 weeks compared to HFD-induced mice), which inhibited obesity by regulating beneficial gut bacteria (*Akermann* and *Bacteroidetes*) and reducing the F/B ratio in suppressing obesity [51]. Research involving hamsters fed a HFD supplemented with 20 g pea protein/100 g diet for 30 days also demonstrated that pea protein consumption could modulate cholesterol levels through the promotion of beneficial gut bacteria growth (*Muribaculaceae* and *Ruminococcaceae*) [10]. Administering adzuki bean protein hydrolysate to HFD-induced obese mice also found a similar regulatory effect on the gut microbiome [88]. Specifically, it decreased harmful gut bacteria (*Mucispirillum*, *Bilophila* and *Peptococcus*) and increased beneficial gut bacteria (*Lactobacillaceae*, *Eisenbergiella* and *Alistipes*), accompanied by weight loss [89].

Preliminary research has found that pulse proteins have a regulatory effect on regulating the gut microbiome, as well as a resulting improvement on MetS. Given the important role of the gut microbiome in regulating human health, more research is needed in the future to confirm the impact of pulse proteins and their hydrolysates in this regard.

## 4. Clinical Studies of Pulse Proteins and Their Hydrolysates on MetS

The results of clinical studies have indicated that the consumption of proteins derived from mung beans and peas has the potential to alleviate risk factors associated with MetS. GLUCODIA^TM^ is a commercially available mung bean protein isolate produced as a by-product during starch production. A study was designed to investigate the effects of GLUCODIA^TM^ on glucose and lipid metabolism. Healthy subjects were randomly assigned to two groups: a control group and a test group (25 people in each group). The test group consumed the assigned candies twice a day before breakfast and dinner (total daily intake of candies containing 12 g GLUCODIA^TM^), while the control group took casein candies in the same way. Compared to the control group, subjects in the test group had significantly lower mean insulin levels, assessed values of the homeostatic model assessment of insulin resistance and mean TG levels as well as significantly higher serum adiponectin levels [90]. The objective of another study was to examine the effects of two doses of yellow pea protein powder (NUTRALYS^®^) on postprandial glycaemic, insulinemic, glucose-dependent insulinotropic polypeptide (GIP) and glucagon-like peptide-1 (GLP-1) response in healthy individuals after ingestion of a high-carbohydrate beverage. Thirty-one participants were randomly assigned to consume 50 g glucose (control), 50 g glucose with 25 g pea protein (Test 1) and 50 g glucose with 50 g pea protein (Test 2) on three separate days. Compared with controls, the glucose incremental Area under the Curve (iAUC180) was significantly lower and insulin iAUC 180 was significantly higher after Test 1 and Test 2, whereas there were no significant differences in GIP and GLP-1 release [17]. 

In addition, the ability of pulse protein hydrolysates to treat MetS and regulate the gut microbiome was investigated. A randomized controlled study examined the effects of high-protein supplements (hydrolysed pea protein) during an energy-restricted diet in obese individuals with MetS [91], showing that visceral fat was significantly reduced. Noticeably, the reducing was closely associated with increased microbial diversity, which suggested that pea proteins may regulate the gut microbiome to treat obesity.

## 5. Conclusions and Outlook

As the information compiled in this review demonstrates, pulse proteins and their hydrolysates show beneficial effects on MetS, including anti-hypertension, anti-hyperglycemia, anti-dyslipidemia and anti-obesity activities, based on in vitro and in vivo findings. The positive effects they ultimately exert on the prevention and treatment of MetS might be closely associated with their regulation effect on the gut microbiome.

However, most studies only discussed the bioactivities of one type of pulse protein and its hydrolysates, and there is a lack of comparing and discussing the bioactivities of multiple pulse proteins and their hydrolysates at similar periods and dose levels. Especially in terms of dosage levels, the differences between studies range from 40 mg/kg to 3200 mg/kg (Table 2), a difference of up to 80 times. That is to say, although existing studies have found that many pulse proteins and their hydrolysates have positive effects on MetS, it cannot be determined which one is better. In the development of ideal foods for MetS, it is necessary to screen for pulse proteins or hydrolysates with superior activity through comparative studies at the same level. In addition, clinical research is also necessary, after preliminary confirmation through in vitro and in vivo studies.

In the treatment of MetS, apart from a small number of studies involving pulse proteins, most studies focused on pulse protein hydrolysates and peptides. In addition to the influence of pulse protein raw materials, the pre-treatment of proteins and the selection of hydrolytic enzymes also significantly affected the activity of protein peptides. There are still many areas worth exploring in the research of functional peptides from pulse proteins. Furthermore, protein and genetic engineering modifications have also been used to improve the MetS regulatory activities of pulse proteins. For instance, lactostatin (a cholesterol-lowering bioactive peptide) was engineered into mung bean 8Sα globulin through protein engineering [92] and another hypocholesterolemic peptide (LPYPR) was engineered into mung bean 8Sα globulin by genetic engineering [93]. 

This review also provides a preliminary understanding of pulse proteins and their hydrolysates in regards to alleviating MetS based on their regulatory effect on the gut microbiome, but as the research is not yet in-depth and widespread, it is still worth exploring whether pulse proteins could be considered a protein-based prebiotic. In addition, the application of pulse proteins and their hydrolysates in the production of functional health foods that are beneficial to patients with MetS should also be part of the direction of subsequent efforts.

## Figures and Tables

**Figure 1 nutrients-16-01845-f001:**
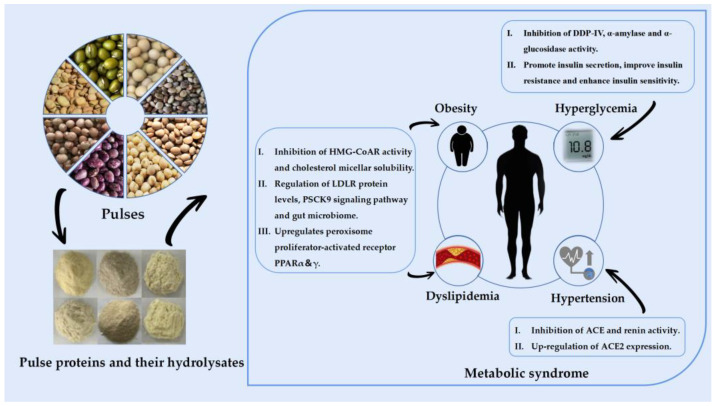
Pathways of pulse proteins and their hydrolysates for improving MetS. ACE: angiotensin-converting enzyme; DDP-Ⅳ: dipeptidyl peptidase IV; HMG-CoAR: 3-hydroxy-3-methylglutaryl CoA reductase; LDLR: low-density lipoprotein receptor; PSCK9: proprotein convertase subtilisin/kexin type 9.

**Figure 2 nutrients-16-01845-f002:**
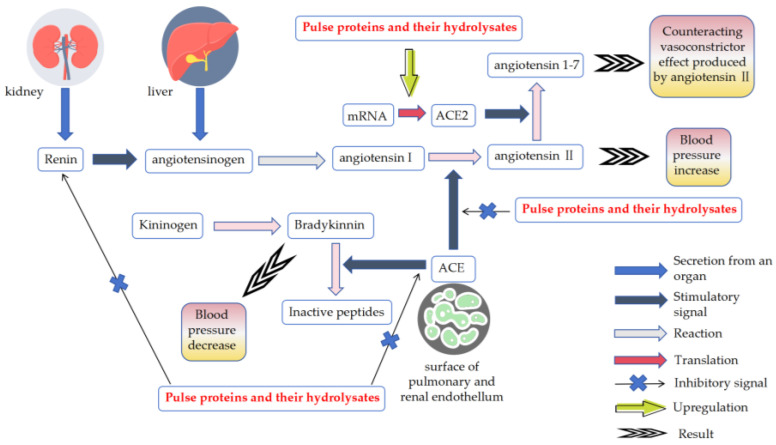
Mechanisms of action of pulse proteins and their hydrolysates. ACE: angiotensin-converting enzyme; ACE2: angiotensin-converting enzyme Ⅱ.

**Figure 3 nutrients-16-01845-f003:**
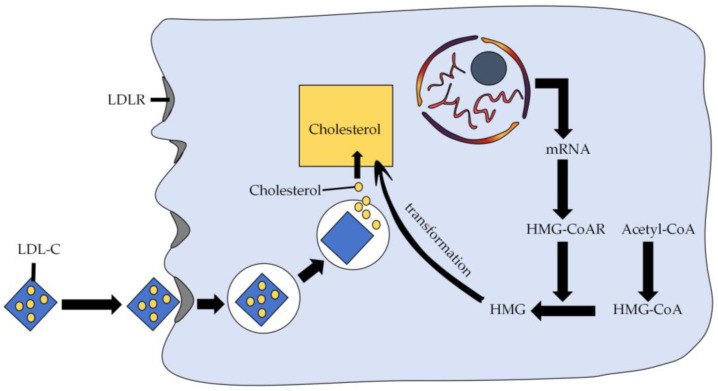
Mechanisms of cholesterol uptake and synthesis by cells. LDL-C: low-density lipoprotein cholesterol; LDLR: low-density lipoprotein receptor; HMG-CoAR: 3-hydroxy-3-methylglutaryl CoA reductase; HMG-CoA: 3-hydroxy-3-methylglutaryl CoA; HMG: 3-hydroxy-3-methylglutaryl; Acetyl-CoA: acetyl coenzyme A.

**Table 1 nutrients-16-01845-t001:** Summary of recent in vitro studies on the health benefits of pulse proteins and their hydrolysates.

Health Benefits	Sources of Pulse Proteins	Protease/Processing Methods	Products/Peptides Sequence	Testing Methods	Key Findings	References
Anti-hypertension	Lupin	Alcalase and flavourzyme	Protein hydrolysates	ACE inhibition	IC_50_: 3.140–3.317 mg/mL	[21]
		Alcalase and flavourzyme	Protein hydrolysates	ACE inhibition	IC_50_: 0.10–0.21 mg/mL	[22]
	Mung bean	Alcalase, neutrase, papain and protamex	Protein hydrolysates	ACE inhibition	IC_50_: 4.66–10.27 μg/mL	[23]
		Bromelain	Protein hydrolysates, LPRL, YADLVE, LRLESF, HLNVVHEN and PGSGCAGTDL	ACE and renin inhibition	IC_50_: 0.004–0.95 mg/mL (ACE); 0.95–1.29 mg/mL (Renin).	[24]
		*Rhizopus* sp. fermentation	Protein extract	ACE inhibition	Inhibition activity: 68.85–75.00%.	[25]
	Pea	Thermoase and pepsin	AKSLSDRFSY, LSDRFS, SDRFSY	Regulation of ACE2 expression in vascular smooth muscle cells	Upregulation of ACE2 expression activity.	[26]
		Commercial pea protein hydrolysates	Commercial pea protein hydrolysates with major peptides identified	ACE inhibition	IC_50_: 0.43–0.61 mg/mL	[27]
	Pigeon pea	Pepsin, trypsin and chymotrypsin	Protein hydrolysates (MW< 3 kDa)	ACE inhibition	IC_50_: 11.76 μg/mL	[28]
		Pepsin and pancreatin	Pigeon pea peptides	ACE inhibition	Inhibition activity: 53.04% (hydrolysed boiled pigeon pea); 71.53% (hydrolysed pigeon pea tempe)	[29]
		Thermoase	Protein hydrolysates	Inhibition of ACE and renin	Inhibition activity: 55.2–78.6% (ACE); 18.6–27.0% (Renin).	[30]
	Chickpea	Alcalase	Chickpea protein concentrate hydrolysates	ACE inhibition	IC_50_: 22.43–47.48 mg/mL	[31]
Anti-hyperglycemia	Lupin	Pepsin and pancreatin	γ-conglutin peptides	DPP-IV inhibition; Caco-2 cells, HepG2 cells and 3T3-L1 cells	Inhibition activity: 100%; Improving insulin receptor sensitivity and inhibiting hepatic gluconeogenesis.	[32]
		Pepsin and pancreatin	γ-conglutin peptides	Inhibition of DPP-IV and α-glucosidase, pancreatic β-cells, myoblasts and primary human skeletal muscle myotubes (HSMM)	DPP-IV inhibition activity but not α-glucosidase inhibition activity; No insulinotropic action in pancreatic β-cells. Possessing strong insulin-mimetic actions.	[33]
		Alcalase and flavourzyme	Protein hydrolysates	Inhibition of α-amylase and α-glucosidase	IC_50_: 1.416–3.153 mg/mL (α-amylase); 1.65–2.08 mg/mL (α-glucosidase).	[21]
		Alcalase and flavourzyme	Protein hydrolysates, SPRRF, FE, RR, RPR, PPGIP and LRP	Inhibition of α-amylase and α-glucosidase	IC_50_:1.66–4.87 mg/mL (α-amylase); 1.65–4.51 mg/mL (α-glucosidase).	[34]
	Pea	Alcalase, pepsin, trypsin and chymotrypsin	Protein-derived peptides	Inhibition of α-amylase and α-glucosidase	Inhibition activity: 31.00% (α-amylase); 53.35% (α-glucosidase).	[35]
		Commercial pea protein hydrolysates	Commercial pea protein hydrolysates with major peptides identified	DPP-IV inhibition	IC_50_: 1.00–1.33 mg/mL	[27]
		—	Protein-derived peptides	AML-12 Cells	Inhibiting glucagon-stimulated hepatic glucose production; Regulating both gluconeogenic and insulin signaling pathways; Inhibiting HGP dependent on the gluconeogenic signaling but not the insulin signaling.	[36]
	Pigeon pea	Thermoase	Protein hydrolysates	Inhibition of α-amylase and α-glucosidase	Inhibition activity: 10.0–27.1% (α-amylase); 20.0–40% (α-glucosidase).	[30]
	Chickpea	Subtilisin and trypsin	Protein hydrolysates	α-glucosidase inhibition	IC_50_: 0.52 mg/mL	[37]
		The enzyme–bacteria synergy method	Protein hydrolysates	α-glucosidase inhibition	Inhibition activity: 32.5–58.22%	[38]
	Cowpea	Pepsin and trypsin	Protein hydrolysates	α-amylase inhibition	IC_50_: 0.127–0.223 mg/mL	[39]
Anti-dyslipidemia	Lupin	Pepsin	YDFYPSSTKDQQS	HMG-CoAR Inhibition, HepG2 cells	Inhibition activity: 11.7–87.4%; Activating SREBP-1.	[40]
		Peptide Synthesis	GQEQSHQDEGVIVR	PCSK9−LDLR binding assay, HepG2 cells	Impairing the protein-protein interaction between PCSK9^D374Y^ and LDLR.	[41]
		Peptide Synthesis	LILPHKSDAD	PCSK9−LDLR binding assay, HepG2 cells	Ameliorating the HepG2 ability to uptake LDL; Increasing the LDLR protein level on the cell surface.	[42]
		Trypsin	GQEQSHQDEGVIVR	HMG-CoAR inhibition, HepG2 cells, Caco-2 cells	IC_50_: 99.5 μM; Cholesterol reduction by regulation of PCSK9^D374Y^ or LDL-R pathways.	[43]
		Peptide Synthesis	LYLPKHSDRD	PCSK9-LDLR binding assay, HepG2 cells	IC_50_: 88.9 μM (HMG-CoAR); 0.7 μM (PCSK9).	[44]
	Mung bean	Thermosonication pre-treatment, pepsin and pancreatin	Protein hydrolysates	Inhibition of cholesterol micellar solubility and HMG-CoAR	Thermosonication pre-treatment can improve the inhibition of cholesterol micellar solubility.	[45]
	Pigeon pea	Pepsin and pancreatin	Protein peptides	Inhibition of cholesterol micellar solubility and HMG-CoAR	Inhibition activity: 41–77% (cholesterol micellar solubility); 11–36% (HMG-CoAR).	[46]
	Chickpea	Alcalase	VFVRN	Inhibition of cholesterol micellar solubility and HMG-CoAR; HepG2 cells	Inhibition activity: 30.24–66.67% (cholesterol micellar solubility); 22.90–64.30% (HMG-CoAR); Decreasing the expressions of SREBP-1c, SREBP-2 and LXR α.	[19]
	Cowpea	Pepsin and pancreatin	Protein hydrolysates (<3 kDa), QDF	HMG-CoAR inhibition	Inhibition activity: 85.8–95.0%	[47]
		Pepsin and pancreatin	IAF, QGF, QDF	HMG-CoAR inhibition	Inhibition activity: 69–78%	[48]
	Faba bean	Thermal processing, pepsin and pancreatin	Protein hydrolysates	Inhibition of cholesterol micellar solubility and HMG-CoAR	Thermal processing can improve the inhibition of cholesterol micellar solubility.	[49]
Anti-obesity	Pea	Pepsin and a pancreatin-bile salts mixture	Vicilin hydrolysates	3T3-L1 cells	Modulating the mRNA expression levels of markers of differentiation and glucose uptake and metabolism in 3T3-L1 adipocytes; Up-regulating the expression levels of PPARγ and exhibiting PPARγ ligand activity.	[50]
		Isolated from pea seeds	Albumin	3T3-L1 cells	Inhibition of lipid accumulation in 3T3-L1 cells.	[51]

ACE: angiotensin-converting enzyme; ACE2: angiotensin-converting enzyme Ⅱ; DDP-Ⅳ: dipeptidyl peptidase IV; HMG-CoAR: 3-hydroxy-3-methylglutaryl CoA reductase; LDLR: low-density lipoprotein receptor; HGP: hepatic glucose production; PCSK9: proprotein convertase subtilisin/kexin type 9; SREBP: sterol regulatory element binding protein; LXR: liver X receptor; PPARγ: peroxisome proliferator-activated receptor γ; Caco-2: human colorectal adenocarcinoma cell; HepG2: human hepatocellular carcinomas; 3T3-L1: mouse embryonic fibroblasts; AML-12: Alpha Mouse Liver 12.

**Table 2 nutrients-16-01845-t002:** Summary of recent in vivo studies on the health benefits of pulse proteins and their hydrolysates.

Health Benefits	Sources of Pulse Proteins	Samples/Peptides Sequence	Experiment Models	Doses and Duration	Key Findings	References
Anti- hypertension	Mung bean	LPRL, YADLVE, LRLESF, LRLESF, LRLESF	Male SHRs	20 mg/kg BW for one oral administration	↓ SBP: 36 mmHg (positive control 15 mmHg), 27 mmHg after 24 h	[24]
	Pigeon pea	Protein hydrolysates	Male SHRs	100 mg/kg BW for one oral administration	↓ SBP: 34.6 mmHg after 6 h, 32.12 mmHg after 24 h	[52]
	Chickpea	Protein hydrolysates	Male SHRs	1.2 g/kg BW for one oral administration	↓ SBP: 29.9~61.9 mmHg after 3~5 h	[53]
Anti- hyperglycemia	Lupin	Lupanine	T2D was induced in a rat blood glucose model	28 mg/kg BW γ-conglutin + 20 mg/kg BW lupanine for 7 days	↓ Glu: Cγ+ Lupanine was similar to the Metformin + Glibenclamide ↓ BW	[54]
		Protein isolate	HFD-induced insulin resistance rats	2 g/kg BW for 3 weeks	↓ Glu: 33% (pre- vs. post-treatment) ↓ AUC	[55]
	Mung bean	Peptides	HFD-induced prediabetes C57BL/6 mice	245 mg/kg for 5 weeks	↓ Glu: (positive control) ↓ INS	[20]
	Pea	Oligopeptide	STZ-induced diabetic mice	3200 mg/kg BW for 4 weeks	↓ INS: 65.07% ↑ OGTT ↓ INS	[56]
		Glycoprotein	STZ-induced diabetic mice	400 mg/kg for 6 weeks	↓ Glu: 6 mmol/L (positive control: 1 mmol/L) ↑ Gene: the expression of insulin receptor substrates IRS-1 and IRS-2	[57]
Anti- dyslipidemia	Lupin	Protein hydrolysates	Western diet-fed ApoE^−/−^ mice	100 mg/kg for 12 weeks	↓ TC, LDL-C, TG ↓ mRNA: expression of CXCL1, P-selectin, CD-36, iNOS	[58]
		Protein hydrolysates	Western diet-fed ApoE^−/−^ mice	100 mg/kg for 12 weeks	↓ Protein levels: HMG-CoAR 92.0 ± 2.6% vs. positive control ↓ SREBP2	[59]
		Protein	Acute hyperlipidemia was induced by injection in male albino rats	A binary mixture of α-CD and LPs at the ratio 1:1 (400 mg/kg and 800 mg/kg) for 3 days	↓ TC, TG, LDL-C	[60]
	Pigeon pea	Protein hydrolysates	HFD-induced rats	300 µM/kg for 2 weeks	↓ TG, TC, LDLR	[61]
	Chickpea	RQSHFANAQP	HFD-induced hyperlipidemic mice	40 mg/kg/day for 4 weeks	↓ TG, TC	[62]
		Peptides	HFD-induced obese rats	200 mg/kg BW for 6 weeks	↓ TC, TG, LDL-C	[19]
	Cowpea	Protein isolate	HFD-induced Sprague–Dawley rats	5% (*w*/*w*) cowpea protein isolate for 6 weeks	↓ TC	[63]
	Pea	Protein hydrolysates	GM-induced Wistar rats	200 mg/kg/day for 28 days	↓ TC, TG	[64]
	Lentil	Protein hydrolysates	Male obese (fa/fa) (O) and lean heterozygous (fa/+) (L) Zucker rats	1 g/kg BW for 8 weeks	↓ Glu, LDL, TC, TG	[65]
Anti- obesity	Mung bean	Protein	HFD-induced male C57BL/6 mice	Specific mung bean protein isolate-containing diets (25%) for 4 weeks	↓ BW, TG ↓ cytoplasmic vacuolation	[66]
	Chickpea	Protein hydrolysates	Metabolic dysfunction C57BL/6J mice	800 mg /kg BW for 16 weeks	↓ BW ↑ Gene: Acsl4, Adipor2	[67]
	Pea	Albumin	HFD-induced male C57BL/6 mice	1.50 g/kg BW for 8 weeks	↓ BW, TG, adipocyte hypertrophy, percentage of fat mass ↓ Protein abundance: FASN, C/EBPα pIRS1(Ser302) and p-IRS1(Ser302)	[51]
		Protein	High-caloric diet-induced obese Sprague–Dawley rats	Specific diet supplemented with the pea protein isolate (25%) for 4 weeks	↓ TC, LDL, TG	[68]

↓: decrease; ↑: increase; SBP: systolic blood pressure; Glu: blood glucose; AUC: area under the plasma concentration-time curve; OGTT: oral glucose tolerance test; INS: insulin; TC: serum total cholesterol; TG: triglyceride; LDL-C: low-density lipoprotein cholesterol; LDLR: low-density lipoprotein receptor; CD-36: cluster of differentiation 36; iNOS: inducible nitric oxide synthase; BW: body weight; HMG-CoAR: 3-hydroxy-3-methylglutaryl CoA reductase; STZ: Streptozotocin; HFD: high-fat diet; GM: gentamicin; CXCL1: chemokine (C-X-C motif) ligand 1; FASN: fatty Acid synthase.

**Table 3 nutrients-16-01845-t003:** Summary of recent studies on the effects of pulse proteins and their hydrolysates on the gut microbiome.

Health Benefits	Sources of Pulse Proteins	Samples	Experiment Models	Doses and Duration	Key Findings	References
Anti-hypertension	Pea	Oligopeptide	Diet-induced hypertension rats	50 mg/mL each for 3 weeks	↓: F/B ratio ↑: *Lactobacillaceae*	[86]
Anti-hyperglycemia	Mung bean	Peptides	HFD-induced male C57BL/6 mice	245 mg/kg BW for 5 weeks	↓: *Firmicutes* and *Bacteroidetes.*	[20]
Anti-dyslipidemia	Lupin	Protein	Female crossbred pigs ((German Landrace × Large White) × Pietrain)	130–150 g/kg diet for 4 weeks	↑: Bacteroidetes and Firmicutes phyla.	[87]
Anti-obesity	Mung bean	Protein	HFD-induced male C57BL/6 mice	Specific mung bean protein isolate-containing diets (25%) for 4 weeks	↑: *Bacteroidetes* ↓*: Firmicutes*	[66]
	Pea	Albumin	HFD-induced male C57BL/6 mice	1.50 g/kg BW for 8 weeks	↓: F/B ratio ↑: Akkermansia and Parabacteroides.	[51]
		Protein	HFD-induced hamsters	HFD + pea protein at 20 g/100 g diet for 30 days	↑: *Muribaculaceae* and *Ruminococcaceae.* ↓: *Erysipelotrichaceae* and *Eubacteriaceae.*	[68]
	Adzuki bean	Protein hydrolysates	HFD-induced male C57BL/6 mice	100 mg/kg BW for 12 weeks	↑: Lactobacillus and SCFA-producing bacteria. ↓: Clostridium_sensu_stricto_1, Romboutsia, Blautia, Mucispirillum, Bilophila and Peptococcus.	[88]
		Protein hydrolysates	HFD-induced male C57BL/6 mice	6% heat-treated adzuki bean protein hydrolysates for 12 weeks	↑: Lactobacillaceae, Eisenbergiella, Alistipes, Parabacteroides, Tannerellaceae, Eubacterium_nodatum_group, Acetatifactor, Rikenellaceae and Odoribacter. ↓: Clostridium_sensu_stricto_1, Romboutsia, Blautia, Mucispirillum, Bilophila and Peptococcus.	[89]

↑: Increased relative abundance; ↓: Decreased relative abundance; F/B: Firmicutes/Bacteroidetes; SCFA: short-chain fatty acids; HFD: high-fat diet.

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
