# Peer review of "Pulse Proteins and Their Hydrolysates: A Comprehensive Review of Their Beneficial Effects on Metabolic Syndrome and the Gut Microbiome"

_nutrients, 2024, doi:10.3390/nu16121845_

Round 1

Reviewer 1 Report

Comments and Suggestions for Authors

The aim of this review is to gain a deeper understanding of the value and efficacy of key pulse proteins and their hydrolysates in the treatment of MetS, thereby encouraging further research into their potential applications.Before it is accepted for publication, the following changes/modifications are suggested to make the manuscript more interesting and self-explanatory,

Specific comments:

1. Keywords in their current state repeat the title and are not useful to increase search results.

2. I recommend that the authors include details of the databases used for data collection/extraction (e.g. Web of Science, Scopus, Google Scholar) and the keywords used during the literature search, as well as the time period of the studies included in the review. This will ensure comprehensive coverage of recent and relevant studies, preferably in the introductory section.

3. Authors should produce figures illustrating the molecular mechanisms induced by pulse proteins and their hydrolysates against the diseases reviewed (hypertension, hyperglycaemia, dyslipidaemia, obesity, dysbiosis) and specific agents. Many readers find visual aids more accessible than textual descriptions or tables.

4. References are not formatted according to journal requirements.

5. I recommend a thorough revision of the manuscript to check for repetition.

Reviewer 2 Report

Comments and Suggestions for Authors

There is considerable interest in the plant proteins as alternative macronutrient especially from the perspective of their nutritional value. Interest also extends to their potential health benefits for humans. This review addresses some of the important aspects of pulse proteins in this regard and should be a useful addition to this growing field. In general article is well organized, and literature reviewed is reasonably recent. There are however number of minor issues (outlined below) which need authors' attention.

Specific comments

Line 80-83: Objective statement is not consistent with content of the review. Review isn't about "treating MetS" rather about evaluation of beneficial effects in relation to MetS. Treatment implies therapeutic effect.  Also, there is little information on the application (such as in functional foods or supplements). Please rephrase.

Line 92 - change 'treating' to 'controlling or managing'

Fig 1 - expand abbr. PSCK 9. 

Authors have reviewed literature in four areas related with metabolic syndrome - hyperglycemia, dyslipidemia, hypertension and gut microbiome, dividing into in vitro and in vivo studies. Description of in vitro studies should include information about methodology. For example, method used to measure ACE inhibitory activity for anti- hypertension effects. Were method employed (21,23,27,8,31) similar and results comparable. Authors make a point about effect of molecular weight on ACE activity (111-1170 quoting IC50 of peptides in ug/mL. IC50 values of similar small MW from same source jumps significantly in mg/mL (line 117). Is it due to how peptides were generated? In next example ACE inhibitory activity is presented as percentage (line129). It is desirable that authors should summarise this information in concise manner.

What does 'proteins personally hydrolysed" mean? (line 119)?

Furthermore, lot of discussion in the invitro section is focused on protein hydrolysates - what about intact proteins? does that mean that intact proteins haven't been studied or bioactivity isn't expected in intact pulse proteins.

It is desirable that in vivo studies (these are only animal studies), authors should describe the trial designs in better detail. For example, description of animal models including relevance of the phnotype for better context and understanding. Similarly, what is phenotype of is the phenotype of ApoE-/- mice? how is this relevant in the study.

I appreciate that focus of this review is only pulse proteins and their hydrolysate, I think it would be useful if authors include examples from proteins and hydrolysates from other sources (both plant and dairy) for similar effects as a comparative data. 

Interestingly most of the discussion is focused on one mechanism (ACE inhibition) for anti-hypertension and HMG CoAR for anti-dyslipidemia. Please comment.

line 346 - it should be 'relative abundance" rather than ' abundance'? typ0 MeyS (Line 348)

When discussing beneficial effect of plant proteins supplemented diets please include information on the trial design (ref 86, 87, 51 and 66) rather than just outcome. It is important to understand the context of outcome measures. 

Clinical studies: This section needs more critical examination of the studies presented. For example, difficult to understand the physiological basis of 'significant' improvement biomarkers by 12 gm mung protein ingestion for eight weeks.  GLUCODIA and NUTRALYS appear to be supplements developed for specific purpose. 

Conclusion and outlook section is well presented.

Comments on the Quality of English Language

English is generally very good. Minor editing would be helpful.

Reviewer 3 Report

Comments and Suggestions for Authors

Dear Authors

I reviewed the Manuscript ID: nutrients- 3027385 -  Pulse proteins and their hydrolysates: a comprehensive review of their beneficial effects on metabolic syndrome and gut microbiome

In this Review the authors underline the importance of the pulses in the human health due to the fact that pulse proteins and their hydrolysates have shown promising results in alleviating metabolic syndrome and modulating gut microbiome. The paper reviews the  anti-hypertension, anti-hyperglycemia, anti-dyslipidemia and anti-obesity activities of pulse proteins and their hydrolysates in recent in vitro and in vivo studies, highlighting their potential in the prevention and treatment of metabolic syndrome.

The scientific quality of the paper is good. This work was well designed and performed with proper methodology. However, I noticed a certain discrepancy in the numbering of bibliographical references in the text.

In the text there are some typing errors:

Line 111, 113, 114, 115, 118, 121, 122:  MWs

Line 147: Mas

Line 151: it is necessary to insert a bibliographic reference

Line 168-171: describe the processing methods with more detail.

Line 221: A high -fat diet (HFD)

Line 348: MetS

Round 2

Reviewer 1 Report

Comments and Suggestions for Authors

The authors have revised the manuscript in accordance with the reviewer's instructions. I have no further comments.